# OpenReview forum: "Towards Inference-time Category-wise Safety Steering for Large Language Models"
_NeurIPS.cc/2024/Workshop/SafeGenAi — SafeGenAi Poster_

### Official Review · Reviewer_HynV · 2024-10-09
**Explored two safety steering mechanism for LLM with good discussions**

**Rating:** 7
**Confidence:** 2

**Review:**

The paper explored the category-specific steering vectors and informative steering vectors for LLM safety steering. I am new to the safety steering field so not sure what is the research norm. The paper provides a good explanation on the two vectors and their implication. The discussion reminds me of some papers I've read from CHI and CSCW. Their approach is more qualitative. Again, this is not my field (both in CS and CHI aspects), but it would be great to see a system coming out that can tie both the qualitative and quantitative findings together. The author has a clear plan for their future work and I look forward to that!

---

### Official Review · Reviewer_d63r · 2024-10-09
**This paper introduces an innovative approach to enhancing LLM safety by using category-specific steering vectors. This method allows for fine-grained control over model outputs, effectively reducing harmful or biases content while maintaining the quality and utility of the generated text. The approach is modular, requiring no retraining, making it highly practical for real-world applications. The dual extraction techniques for steering vectors, both unsupervised and supervised, ensure robust safety interventions without compromising performance. While the paper presents a strong foundation, it could benefit from more detailed comparisons with existing methods and a deeper exploration of limitations, such as scalability and transferability. This work makes a valuable contribution to AI Safety, offering practical and easily implementable solutions to ensure safer LLM deployments.**

**Rating:** 7
**Confidence:** 4

**Review:**

This paper presents a compelling and timely approach to enhancing the safety of large language models (LLMs) through the introduction of category-specific steering vectors. This innovative method aims to provide fine-grained control over LLM outputs, which significantly advances current safety steering techniques. By tailoring steering vectors to specific harm categories, the authors enable targeted interventions in the model's generation process, reducing the risk of harmful or biased content. This category-specific approach not only mitigates risks associated with harmful outputs but also preserves the utility and coherence of the model’s responses, making it highly applicable in sensitive applications. The paper emphasizes the pivotal role of category-specific steering vectors in improving the safety alignment of LLM outputs. By differentiating steering signals based on the nature of the potential harm, the authors enhance the precision of safety interventions, making it possible to handle harmful outputs in a more nuanced way. This targeted steering approach contrasts with generalized safety methods, offering more effective interventions that maintain the high quality of the generated text. The proposed methods for extracting informative steering vectors, whether through unsupervised techniques or guided by external safety classifiers, ensure that the model can generate safe content without sacrificing text quality. Additionally, noise removal through pruning further refines the steering signals, leading to more accurate control over the model’s outputs.

In contrast to many recent inference-time safety steering methods that rely on mechanistic interpretability or representation engineering, the authors propose a training-free methodology that does not require retraining or architectural changes to the LLM. This modular approach is designed for ease of integration into existing frameworks, which is crucial for real-world applications where rapid deployment and adaptability are essential. The paper also emphasizes the need for ongoing evaluation and refinement of steering vectors, ensuring their effectiveness across various models and datasets. These modular and adaptable methods address the need for practical safety solutions that can be quickly implemented without significant overhead. The originality of this work lies in addressing a significant gap in LLM safety by focusing on inference-time steering without requiring additional training data or complex adjustments. The dual approach to steering vector extraction—using both unsupervised methods that capture activation differences between harmful and harmless prompts, and guided approaches that leverage external classifiers—demonstrates both robustness and versatility. This design enables the model to remain flexible and maintain high performance even as it is steered toward safer outputs.

While the paper provides a solid foundation, there are areas for further enhancement. A more detailed comparison with existing safety steering methods, particularly in terms of performance metrics, would be beneficial. Additionally, a deeper exploration of the limitations and potential challenges, such as the scalability of steering vectors across different model architectures and their transferability to various harm categories, would enrich the discussion. Such comparisons and insights would help contextualize the contributions of this work within the broader landscape of LLM safety research. My final comment would be that this paper offers a significant contribution to AI safety by proposing practical solutions that enhance the safety and reliability of LLMs. The clarity of the presentation, the originality of the approach, and the focus on real-world applicability make this research an important addition to the field. With further refinement and comparison to other methods, the authors can strengthen their argument for the efficacy and relevance of category-wise safety steering as a crucial tool for ensuring the safe deployment of LLMs in a wide range of sensitive applications.

---

### Official Review · Reviewer_vk1k · 2024-10-11
**Good paper on an important topic**

**Rating:** 7
**Confidence:** 4

**Review:**

This paper studies inference-time safety steering of LLMs. The key idea behind their approach is the computation of *steering vectors* that correspond to the concept of "harmlessness", using it to intervene on intermediate layers during the process of generation. The proposed method is computationally efficient, in the sense that only a single forward pass is required at inference time.

---

### Official Review · Reviewer_uYDV · 2024-10-12
**The paper demonstrates high scholarly rigor, clarity, and adherence to Neurips standards. However, specific lines lack clarity, and some appear unnecessary. While the paper introduces a novel approach using steering vectors, its originality is hindered by the absence of a standardized mechanism for assigning categories to prompts. The significance of the paper is diminished by its limited introduction of novel ideas and its failure to address the trade-off between reduced harmful intent and decreased output quality. The research paper effectively incorporates elements from prior studies and presents concepts with clarity. However, it falls short in successfully integrating the novel idea it attempted to extrapolate.**

**Rating:** 6
**Confidence:** 2

**Review:**

# Review

**Quality:**

The paper exhibits a high level of scholarly rigor, clarity of expression, and adherence to the NeurIPS quality standards.

**Clarity:**

The authors effectively conveyed a compelling concept with substantial clarity. However, specific lines such as 36-37 and 71-72 exhibit a lack of clarity, and certain lines, such as line 86, appear superfluous. Nonetheless, the overall paper scores favorably in terms of clarity.

**Originality:**

The paper presents a novel approach in the field of security by employing steering vectors. While certain ideas appear to be influenced by existing literature, the paper distinguishes itself by introducing a novel concept: category-based steering vectors. However, it lacks a defined methodology for assigning categories to prompts. The broad nature of categories and the absence of a clear mechanism to assign items to specific categories pose challenges. Currently, categories are derived from underlying datasets, but each dataset adopts its own definition of categories. Therefore, a standardized mechanism for category definition is essential. So the paper ranks low in terms of originality.

**Significance:**

The paper presents a novel approach to guiding language models toward safer behavior at inference time, addressing a critical issue in the field of natural language processing. While the proposed method demonstrates a notable reduction in harmful intent, it comes with a corresponding decrease in output quality. The limited introduction of novel ideas and the failure to address this trade-off in a meaningful way result in a lower significance rating for this paper.

**Pros:**

* The paper incorporates elements from other studies that have demonstrated promising results previously.
* The concepts are presented in a clear and comprehensible manner.
* Since the paper deals with an inference time technique, it enables the method to overcome certain limitations associated with training time.

**Cons:**

* The methodology employed in the construction of the steering vector is excessively expansive. To a limited degree, pruned activations address this deficiency.
* There is a substantial decline in both quality and coherence.
* A significant amount of computation is necessary, and the method used to construct the steering vectors is rudimentary and suboptimal.
* The paper proposes category-wise steering but does not elaborate on the process of category construction.